# Service user perspectives and experiences of their diagnosis of psychotic major depression: A qualitative study

Emilia May Loane[1], Norha Vera San Juan[2,3], Allan H. Young[1], Oliver Gale-Grant[1], Margaret Heslin[1]*

1 Institute of Psychiatry, Psychology & Neuroscience, King's College London, London, United Kingdom, 2 Institute for Global Health, University College London, London, United Kingdom, 3 Rapid Research Evaluation and Appraisal Lab (RREAL), University College London, London, United Kingdom

* Margaret.heslin@kcl.ac.uk

## Abstract

### Background

Evidence suggests psychotic major depression can be overlooked in clinical settings and thus lead to delays in diagnosis. There have been multiple theories about why this happens, however no research has investigated this from a service users' perspective.

### Aims

The aim of this study was to explore service users' perspectives and experiences of their diagnosis of psychotic major depression. Where there appeared to be a substantial delay between initial symptoms and diagnosis, reasons for this was explored in depth.

### Methods

This study used a qualitative approach based on semi-structured interviews with service users diagnosed with psychotic major depression. Interviews were audio recorded and transcribed verbatim. Data was analysed using thematic analysis.

### Results

Ten interviews were conducted. Four overarching themes were identified: difficulty retelling the story, barriers to symptom identification, experiences following disclosure, responses to diagnosis. The theme of barriers to symptom identification highlighted that psychotic symptoms can be overlooked in the diagnostic process. Service users reported that health professionals do not always check for psychotic symptoms in clinical assessments. Service users have difficulty verbalising their symptoms, or

**Data availability statement:** Due to conditions of participant consent, data cannot be made openly available but may be shared with researchers on request. To request access, please email research.data@kcl.ac.uk DOI: https://doi.org/10.18742/26003764

**Funding:** The author(s) received no specific funding for this work.

**Competing interests:** MH reports funding from NIHR. AY reports funding from Flow Neuroscience, Novartis, Roche, Janssen, Takeda, Noema pharma, Compass, Astrazenaca, Boehringer Ingelheim, Eli Lilly, LivaNova, Lundbeck, Sunovion, Servier, Livanova, Janssen, Allegan, Bionomics, Sumitomo Dainippon Pharma, Sage, Neurocentrx, NIMH, CIHR, NARSAD, Stanley Medical Research Institute, MRC, Wellcome Trust, Royal College of Physicians, BMA, MSFHR, EU Horizon 2020, NIHR. No other authors reported competing interests.

find it difficult to disclose due to stigma, fear or shame. Short/rushed appointments, lack of consistency with health professionals, being moved between services and medicalising language made disclosure of psychotic symptoms less likely.

## Conclusions

Psychotic major depression should be actively considered as a differential diagnosis by healthcare professionals when assessing an individual for unipolar depression. Healthcare professionals should be mindful of the specific barriers to disclosure of psychotic symptoms, and building rapport should be prioritised to facilitate disclosure.

## Introduction

Psychotic Major Depression (PMD) is defined in ICD-11 as depressive disorders with psychotic symptoms, in which delusions or hallucinations are present during the depressive episodes [1]. Few studies have investigated long-term outcomes in people with PMD. Those that have, highlighted increased risk of mortality compared to other groups with psychosis [2], and an increased risk of attempted or completed suicide compared to other groups with psychosis [2–4]. A meta-analysis demonstrated that the presence of psychosis during a major depressive episode doubles the risk of a suicide attempt in the acute phase and lifetime of the disorder [5] compared to people with major depression without psychotic symptoms (non-PMD).

Due to the potential poor outcomes described above, timely identification and treatment of PMD is vital. Delayed diagnosis means people will stay sicker for longer and thus be more at risk of these negative outcomes. Delayed diagnosis also leads to sub optimal pharmacotherapy, prolonged recovery time [6] and morbidity [7]. Additionally, delays in initiating antipsychotic treatment are associated with worse patient outcomes [8]. However, late diagnosis of PMD is common [9] although there is no agreement on why this is. Some researchers have suggested this is due to clinician's approaches. Rothschild [7] suggest that clinicians miss the psychotic symptoms, focusing instead on the mood symptoms. Rothschild [7] also suggests symptoms may be subtle and thus not identifiable to the mental health professionals. Schatzberg [10] has shown that the diagnosis of PMD requires an extensive examination of the patient, which is not always possible.

Other researchers suggest PMD is diagnosis is delayed in part due to patient factors. Hales [11] suggests that because patients recognise that their thought patterns are unusual, they may attempt to conceal their symptoms from others. This is related to Zalpuri et al's [12] theory that it may be paranoia symptoms which cause concealment. Alternatively, patients may not be well enough to fully articulate their psychotic symptoms [5] or may not be able to articulate the psychotic experiences due to cognitive impairment [13]. Lastly, some patients may not recall psychotic experiences due to their mental state at the time or may minimise their symptoms [14]. Although there are multiple theories as to why PMD is diagnosed late, this has not been fully investigated.

## Aim

This study aimed to explore service users' perspectives and experiences of their diagnosis of PMD. Where there appeared to be a substantial delay between initial symptoms and diagnosis, reasons for this was explored in depth.

## Materials and methods

We followed COREQ reporting guidelines for qualitative research [15]

### Design

This study used qualitative methods to explore the diagnostic process of PMD from the service users' subjective views, experiences, and opinions. Semi-structured one-off interviews were conducted with people who had a formal diagnosis of PMD according to medical records.

### Participants

Purposive sampling [16] was used to recruit a sample of service users between 18–64, who did not need a translator to participate, who had a formal recorded clinical diagnosis of PMD (based on ICD-10 [17] as this was being used in clinical practice at the time of diagnosis) in the last 3 years, and did not have a different psychosis related diagnosis since then.

### Recruitment

Participants were recruited using two methods: consent for contact (C4C); and referrals from clinical teams. C4C was conducted through a research register held by the NHR Maudsley biomedical research centre (BRC) partnered with the south London and Maudsley NHS foundation trust (SLaM). Participants are added to the C4C register when in the care of SLaM and having consented to being contacted regarding research. The study's inclusion and exclusion criteria were used to complete a C4C search, creating a list of potential participants. Eligible participants were then contacted.

Relevant clinical teams within SLAM were contacted with details of the study and asked to refer any potentially relevant service users who were interested in receiving more information about the study. Any referrals were contact directly.

All potentially relevant service users were contacted by phone and given brief information about the study. If the person was interested, they were sent a participant information sheet and given at least 24 hours to consider before being recontacted by the researcher. Verbal informed consent was obtained from each service user after a full explanation of the participant information sheet. The audio recording of informed consent was listened to by another researcher who then signed the consent form to confirm consent had been given.

### Data collection

Due to the COVID-19 pandemic, interviews were conducted through Microsoft teams via videoconferencing or freephone options. Following the recruitment and consent procedure, interviews were conducted, audio or video recorded with the video removed post-interview. No field notes were made. We requested that service users made sure they were in a quiet space and alone during interviews. Interviews lasted approximately 1 hour. Data collection started in 13/06/2022 and ended on 10/08/2023.

Open-ended, in-depth questions were used to reduce bias and increase rapport [18]. This was facilitated by the use of a topic guide. The questions explored the individual's journey to diagnosis, factors that helped and hindered diagnosis and their interactions with mental health services. A debrief sheet and £20 Amazon voucher was emailed post-interview.

Transcription was completed verbatim within the Microsoft teams' transcription tool, which was then confirmed and anonymised by the researcher. No field notes were made. Transcripts were not returned to service users for comment.

## Ethics

Ethical approval was provided by Oxford REC and has received HRA and HCRW approval – 287774. Participants provided verbal informed consent.

## Data analysis

Data were analysed using reflexive thematic analysis [19]. Data was coded using NVivo 14 [20]. The six thematic analysis steps were followed [21]: (1) familiarisation with the data, through repeated reading & annotation; (2) generation of initial codes, through intense open-coding of data to generate an initial coding frame based on thematic categories rooted in the data; (3) identification of themes, through a detailed review of the coding frame to sort codes in to potential themes; (4) review of themes, through refinement of the developing themes; (5) definition & refinement of themes, through exploration of relationships within & between codes, & revision of thematic definitions, and (6) writing up the study findings. Steps 1–3 were performed independently by two authors (EL & MH) who then performed steps 4–6 collaboratively with NVSJ. Data saturation was not considered. Transcripts were not returned to participants for comment. Participants did not provide feedback on the findings.

## Reflexivity

EL conducted all interviews. EL is a female, White British Masters' student with a BSc in Psychology. In preparation for this study, EL completed training in: good clinical practice; gaining informed consent; safeguarding; GDPR; information governance; Nvivo; C4C guidelines; completed a master's module in qualitative research; and took part in several role play interviews. EL previous conducted research interviews for her BSc. EL was responsible for approaching and recruiting all service users. NVSJ is a female, Latina of mixed ethnic background, Senior Research Fellow. MH is a White British female Senior Lecturer.

Short discussions with EL about participation were the only contact participants had prior to interviews in which a rapport/ relationship could be established. The participants knew the research was being conducted by EL as part of her Master's dissertation. EL had limited knowledge about the specific topic when she began the study. As described above, efforts were made to ask open questions so as not to lead participants in their answers or bias responses.

During analysis, MH and NVSJ had reflective sessions where we discussed emerging themes, and different interpretations of the data.

# Results

## Sample

A total of 10 people participated in the interviews. Twenty-eight people declined participation or did not respond to contact. Reasons were not recorded. Participant characteristics are reported in Table 1. The sample included six females, two males, and two service users who identified as non-binary. The age range was 22–56, with six being under 40. Seven service users self-identified as White British and three as Black African. Time from PMD diagnosis was 1–3 years for one service user and over three years for the remaining nine service users.

## Results of the thematic analysis

Four themes emerged from the data and are describe in Table 2: 1) Difficulty retelling the story; 2) Barriers to symptom identification; 3) Experiences following disclosure; 4) Responses to diagnosis.

## Theme 1: Difficulty retelling the story

Service users found it difficult to describe their pathway to diagnosis and their experiences of this time. This topic came up early in many interviews, and was due to a number of factors described below.

**Table 1. Participant characteristics.**

| Characteristics | Number (percentage) |
|---|---|
| Gender | |
| Male | 2 (20%) |
| Female | 6 (60%) |
| Non-binary | 2 (20%) |
| Age | |
| <40 | 6 (60%) |
| >40 | 4 (40%) |
| Ethnicity | |
| White British | 7 (70%) |
| Black African | 3 (30%) |
| Time since PMD diagnosis | |
| < 1 year | 0 (0%) |
| 1-3 years | 1 (10%) |
| >3 years | 9 (90%) |

**Table 2. Themes and sub-themes.**

| Theme | Sub-theme |
|---|---|
| Difficulty retelling the story | *PMD diagnosis not disclosed to service users' <br> *Service user memory issues around time of first contact with services <br> *Comorbidities/ past psychiatric history |
| Symptom identification | *Symptom presentation <br> *Health professionals not asking about psychotic symptoms <br> *Service users having to try to raise psychotic symptoms <br> *Service users needing direct questioning <br> *Service users difficulties with disclosure <br> *Difficulty verbalising <br> *Stigma, fear and shame <br> *Service- and professional related barriers to disclosure <br> *Factors that facilitated disclosure |
| Experiences following disclosure | *Normalising and understanding <br> *Disbelief/denial <br> *Lack of clarity around the significance of psychotic symptoms <br> *Avoidance of labels <br> *Lack of disclosure or explanation of diagnosis |
| Responses to diagnosis | *Response to the PMD diagnosis <br> *Service user frustration with diagnostic instability and lack of certainty |

**PMD diagnosis not disclosed to service users.** Some service users were not told of their diagnosis, this made it difficult to discuss the process of diagnosis because they were not sure when they had received it and so could not discuss the events around the time. For others, they knew their diagnosis of PMD, but that they did not know what it meant, it had never been explained to them.

*"The thing is I for me I only remember being diagnosed with depression, I don't really know where the psychotic bit came in. It was something I was gonna ask actually when I see the psychiatrist." (Service user 1)*

**Service user memory issues around time of first contact with services.** Some service users reported difficulty remembering what happened around the time of first contact with services, or around the time of first diagnosis of PMD. The difficulty remembering related to different aspects of the service users experience, including who they had contact with, what the main problems were that the service user approached services for, what was asked, and what happened. For some service users, this was because the first diagnosis happened a long time before the interview, for others, this seemed to relate to the impact of their mental health on their memory.

*"…anyway, I know, I don't know what happened. I just find myself in mental hospital…" (Service user 2)*

**Comorbidities/ past psychiatric history.** Some service users had a sudden onset of mental health difficulties for which they sought help which made the timeline of experiences, help seeking and diagnosis clear. However, other service users had a long history of mental health difficulties with different diagnoses where it was not clear to the service user what started when.

### Theme 2: Barriers to symptom identification

Salient to the research question, a common theme that was around symptom identification. This included the sub-themes of: Health professionals not asking about psychotic symptoms; Service users difficulties with disclosure; Service- and professional related barriers to disclosure; and Keys to disclosure.

**Symptom presentation.** It was common for service users to report they had approached services with depression and/or anxiety, and sometimes people present to services complaining of non-specific feelings of begin unwell, while not mentioning psychotic symptoms.

*"[Felt] Very tired, because I had, I wasn't sleeping, um, and had low mood, and very anxious, very depressed " (Service user 1)*

*"I just didn't feel normal, I didn't, I didn't understand myself, I didn't understand what was going on, I didn't understand anything really" (Service user 8)*

**Health professionals not asking about psychotic symptoms.** When exploring experiences of early contacts with health and mental health professionals, service users often describe a focus on mood symptoms and a lack of questions on psychotic symptoms.

*"But when I originally went in, they didn't ask me about, the, they didn't directly ask me if I hallucinated. They did directly ask me if I, if I was suicidal, or if I hurt myself, if I'd ever tried to commit suicide. That they've asked me a lot of direct questions about my depression. They didn't ask me so many direct questions about my psychosis." (Service user 5)*

As a result of this, some service users reported having to raise the issue of psychotic symptoms themselves. Following raising the issue themselves, this led to more regular enquiry from health professionals.

*"After I said that I had upsetting and unusual experiences, they asked about them regularly. About how I was doing with them, but I was the first person to mention it. They didn't ask me first." (Service user 5)*

Some service users said that they needed direct questioning to be able to confirm that they were seeing or hearing things that other people could not see and hear. That they were unable to voice these issues voluntarily. This is related to the sub theme of service users difficulties with disclosure, described below.

**Service user difficulties with disclosure.**  Even when psychotic symptoms were asked about by health professionals, some service users found it difficult to disclose for a number of different reasons.

*Difficulty verbalising:*    Some service users found it difficult to verbalise what they were experiencing. This was sometimes related to not having the language to describe what was going on, but was also related to issues with thinking clearly.

*"I found it very hard, I couldn't really express myself properly…um, I think I was very agitated, and that was affecting my normal course of thoughts sort of thing" (Service user 4)*

*Stigma, fear and shame:*    Lack of disclosure related to fear, shame or stigma were common. Service users openly referenced stigma and taboo around their experiences.

*"Stigma, um, stigma. Basically you know, um, people stigmatising, and you, you're not well, you're, you're strange… Well, you're strange… I did feel there was a taboo. Maybe that was unconscious cause I, I, was having major hallucinations." (Service user 6)*

Related to this was feeling shame about the symptoms.

*"so I I I've felt ashamed of that I was hallucinating." (Service user 5)*

Service users feared that MHP would not believe them if they shared their experiences. However, this also linked into a fear of being hospitalised if they disclosed their experiences.

*"It was just too weird, I thought, for them to comprehend, Or… they might try and shut me away or something." (Service user 4)*

**Service- and professional related barriers to disclosure.**  There were a number of barriers to disclosure raised by service users that were related to services and health professionals themselves. Short appointment slots made it difficult to build rapport and for service users' to have the time and space to express themselves. Similarly, health professionals rushing appointments or reacting to service users in a negative way made service users feel uncomfortable and reduced the likelihood of them disclosing the full picture of what they were experiencing.

*"I was trying to think, I was trying to answer a question, but it, I was, I was not doing it fast enough pace, and, and he was getting more frustrated, and, and the more frustrated (he) was getting, the more confused I was feeling, so it would, it did not help at all. At the time I remember that I was feeling more stressed" (Service user 7)*

The use of clinical terms like hallucination and delusion was unhelpful to service users. This felt medicalising to service users making them feel uncomfortable in opening up more. Rather they preferred health professionals to talk about their experiences as experiences, not clinical symptoms:

*" I feel like when they talk to me, like so when they took me seriously, and talked at my level, like, yes everything I'm going through is, is real for me, therefore it's a real thing and we can talk about it like that" (Service user 5)*

A lack of consistency with staff also made it difficult for service users to build up a rapport with staff and left them feeling frustrated:

*" …the same nurse, because it wasn't always the same… Sometimes I might have one or two the same, then the next time it someone different, and I'm like I have to explain the same thing again to you. This is frustrating. That's why I came through… How many times do I have to say the same thing over and over and over and over again? Which wound me up" (Service user 9)*

Similarly, being sent between services was seen to be particularly unhelpful:

*"I was thrown between one team and another, because they the first team was like ohh you're not psychotic, you have personality disorder, and then they were like no you've autistic, and then they refer to the autism clinic, and then I was sitting with this, on this waiting list to get seen by the autism clinic for ages" (Service user 5)*

**Factors that facilitated disclosure.** A key to disclosure seemed to be MHP building trust by taking an understanding and empathetic listening stance, and really listening to service users while being patient.

*"Just, I think, being able to talk to someone and knowing that I, that they, they're taking me seriously, and they don't think I'm a weirdo, that's very helpful, it made me trust people more, it made me trust my doctors more, and made me open up more" (Service user 5)*

These positive experiences lead to increased trust in MHP and disclosure, which helps with a timely diagnosis and treatment. Suggesting, how the practitioner interacts with the patient predicts their outcome and is the key to accurate diagnosis.

**Theme 3: Experiences following disclosure**

Once service users had disclosed psychotic symptoms, there were a number of different types of responses from health professionals that had different impacts on the service user. These include normalisation, disbelief/denial, lack of clarity around the significance, avoidance of labels, and explanation of diagnosis.

**Normalisation and understanding.** When mental health professionals normalised experiences, it helped service users feel less stressed, and less isolated, and had a positive impact of the way the person felt about their mental health and their experiences.

*"… the doctors at (hospital) made me feel like I had just a cold and everything I was experiencing was perfectly understandable and not normal but normal for a lot of people." (Service user 5)*

This allowed service users to open up more about their experiences and symptoms.

**Disbelief/denial.** Some service users described that when they did raise psychotic symptoms with mental health professionals, they were not always believed or taken seriously.

*"oh, you're not hearing voices. Your head isn't loud. You're not. You're fine… Yeah, but they entirely brushed it all off" (Service user 8)*

This made service users feel angry and dislike towards the services.

**Lack of clarity around the significance of psychotic symptoms.** Sometime health professionals were unsure about the significance of psychotic symptoms. There was sometimes a lack of clarity around whether this was part of the clinical picture, or whether this was an unrelated experience that is common in the general population.

*"Because at first I was just hearing voices, and he was, quite a lot of people just hear voices, anyway, she wasn't too sure,it wasn't until I started getting like, paranoia, I was like, thinking things like the TV was talking to me, stuff like that. At that point he was like OK, that's a bit more than just hearing voices" (Service user 3)*

There was also some discussion around what was causing the psychotic symptoms, and what this meant for the clinical picture. Service users did not talk about these issues of clinical significance in a negative way but rather just reflecting on the process and conversations that were happening on the way to diagnosis.

**Avoidance of labels.** Some service users described their mental health team not wanting to label them with a diagnosis, but just treat the symptoms.

*"We did have a conversation about it, and they originally said that, um, it, it was probably best not to put a label on me because why would I, Why do I need a label, It would just make me feel more ill, um, and like, make me try to conform to that label" (Service user 5)*

This strongly relates to the sub-theme identified earlier of PMD diagnosis not being disclosed to service users. A possibility is that health professionals are purposively not giving patients a diagnosis, or at least a specific diagnosis with the word psychosis in it, so that any negative consequences of this label can be avoided. Related to this is the lack of disclosure or explanation of diagnosis described below.

**Lack of disclosure or explanation of diagnosis.** Although already discussed within theme 1 because it led to difficulties retelling the story, lack of disclosure or explanation of diagnosis is also important to discuss in relation to service user experience. Many service users expressed frustration around the issue mentioned earlier of not being informed of the diagnosis or of having it explained as they felt it would have helped their understanding of themselves, and could have aided with their recovery.

*"um, I think it's a bit of a missing piece of the jigsaw it might. That might well have really helped… um, I just I think if they I I if someone explained what psychosis meant along the way a lot earlier along the way, it would have helped a lot." (Service user 2)*

In fact, some service users suggested that clearly explaining the diagnosis should happen for others:

*"And the consultant could have been more verbal with me, explaining, taking time to explain what was happening to me…I think it would have made sense in my mind which would have put me more at rest" (Service user 4)*

*"In the future if someone was getting a diagnosis of psychotic major depression, or psychosis and depression, I think it would be helpful if that was explained to the person, like what is this and what does that actually mean, and what does that mean treatment wise, so that they wouldn't go through what I did, of being like I'm crazy and I'm sick and I'm not gonna be able to get better" (Service user 5)*

### Theme 4: Responses to diagnosis

**Response to the PMD diagnosis.** Some service users broadly agreed with the PMD diagnosis and even found it helpful.

*"…As I got older, the, the diagnosis of psychosis at least help me to understand myself better, if that makes sense, because I was like this is why this is happening to me, because I have psychosis, and this is how I can handle it, I know that the things I'm experiencing aren't really real. Therefore I can try and not let them hurt me as much" (Service user 5)*

However, others found it a very negative and distressing experience.

*"They did actually put on my, um, um, diagnosis that went back to the GP, um, psychotic, um, psychotic something or other and, um, I asked them to actually take it off, because I thought it sounded terrible, and, and, you know in the wrong hands, I've felt that that could go against me… um, because people Freeze up on that particular word, don't they, psychotic, they think it means split personality and things like that… (I felt) like a great weight, of, being put on me, rather than the great weight lifted off… I thought I was never, I never, gonna be able to be free again to do things I wanted to do, um, and that people would never see me the same, I asked them to take it off my notes, which was, um, about a year after"* (Service user 4)

Other service users found receiving a PMD diagnosis both positive and negative.

*"Umm I guess it was, I want to say positive, because I didn't, I could have got more ill without, like, help. So I'm glad I got the diagnosis, cause then I wouldn't been put on the medication, but yeah, I would say in the beginning it was very negative and hard to adjust to"* (Service user 3)

**Service user frustration with diagnostic instability and lack of certainty.** Within the interviews, the topic of diagnostic instability was naturally brought up by the service users without directly questioning the accuracy of their diagnosis. Diagnostic instability was explained using a description of the service user's diagnostic history, which included a display of the change in theoretical and actual diagnoses given to the patient.

*"Yeah, so, from when I was 17 to 23, I, I started out with a diagnosis of depression and uh, anxiety, and non-organic psychosis. Then they said personality disorder. Then they said autism. Then they said paranoid schizophrenia. Then they said um, definitely psychosis of some sort, and now I have the, the, and anxiety, and now I have an official diagnosis of OCD with psychotic symptoms."* (Service user 5)

This led to clear frustration with the diagnostic process

*"…then they slapped on the diagnosis Major depression."* (Service user 5)

The diagnostic inconsistency seen in PMD causes the patient to have negative feelings towards their diagnosis.

*"A lot of them made me feel, they just made me feel a bit icky. Though I was like, Great So I I've gone. It's like being thrown from the pan into the fire. It's like I've gone from one icky diagnosis to something that's even worse"* (Service user 5)

## Discussion

This study aimed to explore service users' perspectives and experiences of their diagnosis of psychotic major depression, and where there appeared to be a substantial delay between initial symptoms and diagnosis, explore this in more depth. Difficulty retelling the story (theme 1) was common and made it difficult to get full details about the diagnostic process. This theme included some service users not knowing that they had a diagnosis of PMD, confusion around when the diagnosis of PMD had been given and service user memory issues. This was also evident during recruitment when some potential service users declined participation on the basis that they did not have a diagnosis of PMD even though it was recorded as such in clinical records. As well as impacting this study, this has implications for future work in terms of obtaining a reliable narrative, and recruitment to related studies.

Despite issues around retelling the story, a major theme was identified which related to late diagnosis – barriers to symptom identification. This highlighted that there are multiple reasons why symptoms are not identified. Service users often approached services with "depression and/or anxiety", and therefore, it is not surprising that health professionals did not elicit psychotic phenomena via interview. Mood being the main topic discussed opens the possibility for a solely mood-based diagnosis. Even when psychotic symptoms were enquired about some service users had difficulty verbalising their symptoms, and others found it difficult to disclose due to stigma, fear or shame. Service level and professional related issues which made it less likely service users would disclose included short or rushed appointments, lack of consistency with health professionals, being moved between services and medicalising language. Factors that facilitated disclosure were normalising symptoms, and an understanding and empathetic listening stance from health care professionals.

Service users also described their experiences following disclosure of psychotic symptoms. Some clinical staff responses to disclosure that were helpful, such as normalising the experiences, and some that were unhelpful, such as not believing the service user were having the experiences they described or not taking the disclosure seriously. As well as impacting on the service users' wellbeing, these experiences are likely to impact on future disclosures so have consequences for future diagnostic accuracy.

## Strengths and limitations

This study was the first to try to explore service users' perspectives and experiences of their diagnosis of psychotic major depression, and where there appeared to be a substantial delay between initial symptoms and diagnosis, explore this in more depth. The method used included open ended questions about the diagnostic process so as not to lead the service users answers in a particular way. Although the sample size was small, long and extensive interviews were conducted, which provided a large amount of in-depth data. However, the sample was somewhat homogenous with only White British and Black African service users. A more diverse population in future studies would allow for a more generalisable picture of the pathway to diagnosis of PMD. Following the COREQ also strengthens this paper.

Difficulty with retelling the story as discussed above also has implications of the quality of the data. Further, service users' retrospective accounts are subject to distortions and recall bias, especially when the majority of service users had been diagnosed more than 3 years before the interview. In relation to this, cognitive issues and confusion (as mentioned above) at the time of presentation may have led to poor recall of events. Additionally, due to the stigma and shame around psychotic symptoms, there may have been some minimisation and avoidance of disclosure. Especially because the interviewer was a young white British female, there may be a mismatch in life experiences and a lack of trust. Moreover, the interviews were over Microsoft teams; this was an uncontrollable factor due to the Covid-19 pandemic. Interviewing via video/call affects the rapport generated and is often criticised [22], although there is an argument to be made as to whether such a personal experience surrounded by stigma and shame is easier to discuss over a video/call. Additionally, there is a lack of research showing that virtual interviews cause worse outcomes [23]. Finally, during recruitment only access to diagnosis and contact details of potential participants was available. Therefore, a comparison of the clinical differences between those who did and did not agree to participate was not possible. Similarly, lack of access to clinical notes means exploration of clinician's difficulty in generating a diagnosis, or the patient's understanding of their diagnosis as assessed by the clinical team was not possible.

## Clinical implications

There are several clinical implications to this work. Firstly, it is vital that health professionals ask about psychotic symptoms in a thorough and proactive manner in service users who present with depression and/or anxiety. A failure to elicit psychotic phenomena means PMD and other psychotic diagnoses may be missed, resulting in a longer duration of untreated psychosis. Additionally, the approach adopted by health professionals with service users is essential for

disclosure. An understanding and normalising stance when working with patients increases disclosure, leading to a more accurate diagnosis. Implementing non-judgmental active listening, understanding and compassion into clinical practice is vital. Lastly, lack of disclosure and explanation of PMD diagnosis has a negative impact on service users and should be reflected on in clinical practice.

### Future research

Our finding that psychosis was not always proactively elicited, and the fact that several participants had delayed diagnoses, suggests that there may be a number of people in whom potential PMD diagnoses are being missed. This likely results in individuals who are psychotic being treated for depression. Future research should explore what proportion of people living with depression would meet the diagnostic criteria for PMD, and hence benefit from a different treatment approach.

### Conclusions

Psychotic major depression should be actively considered as a differential diagnosis by healthcare professionals when assessing an individual for unipolar depression. Healthcare professionals should be mindful of the specific barriers to disclosure of psychotic symptoms, and building rapport should be prioritised to facilitate disclosure. Not disclosing or explaining a PMD diagnosis has a negative impact on service user wellbeing and should be avoided.

### Acknowledgments

This paper represents independent research funded by the National Institute for Health and Care Research (NIHR) Biomedical Research Centre at South London and Maudsley NHS Foundation Trust in partnership with King's College London. The views expressed are those of the author(s) and not necessarily those of the NHS, the NIHR or the Department of Health and Social Care. We acknowledge and are very grateful for the support of the Maudsley Biomedical Research Centre's Young Person's Mental Health Advisory Group, and the Maudsley Biomedical Research Centre's Service User Advisory Group.

Professor Young's independent research is funded by the National Institute for Health and Care Research (NIHR) Maudsley Biomedical Research Centre at South London and Maudsley NHS Foundation Trust and King's College London. The views expressed are those of the author(s) and not necessarily those of the NIHR or the Department of Health and Social Care

### Author contributions

**Conceptualization:** Norha Vera San Juan, Allan H Young, Oliver Gale-Grant, Margaret Heslin.

**Data curation:** Emilia May Loane, Margaret Heslin.

**Formal analysis:** Emilia May Loane, Norha Vera San Juan, Margaret Heslin.

**Investigation:** Emilia May Loane, Margaret Heslin.

**Methodology:** Emilia May Loane, Norha Vera San Juan, Margaret Heslin.

**Project administration:** Emilia May Loane, Margaret Heslin.

**Supervision:** Margaret Heslin.

**Validation:** Margaret Heslin.

**Writing – original draft:** Emilia May Loane.

**Writing – review & editing:** Norha Vera San Juan, Allan H Young, Oliver Gale-Grant, Margaret Heslin.

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
