## [Decision Letter · Decision Letter 0]

7 Jan 2025

PONE-D-24-19547Service user perspectives and experiences of their diagnosis of psychotic major depression: a qualitative studyPLOS ONE

Dear Dr. Heslin,

Thank you for submitting your manuscript to PLOS ONE. After careful consideration, we feel that it has merit but does not fully meet PLOS ONE’s publication criteria as it currently stands. Therefore, we invite you to submit a revised version of the manuscript that addresses the points raised during the review process.

**The manuscript has potential but requires substantial revision to address all points raised by the reviewers. The authors must:**

**Respond comprehensively to all reviewer comments, ensuring clarity on recruitment methods, diagnostic criteria, study population, and sampling strategies.****Adhere to the COREQ checklist (or an equivalent) and upload the completed checklist within your response. This should ensure thorough reporting of objectives, sampling strategies, data collection, analysis, and discussions of bias and limitations.****Clarify conflicting points raised by the reviewers, particularly around diagnostic definitions, participant recruitment, and how the diagnosis was ultimately made.**

**The revisions should also address minor stylistic or structural issues mentioned in the reviews to enhance clarity and coherence.**

We look forward to receiving your revised manuscript.

Kind regards,

Sarah Liddle

Academic Editor

PLOS ONE

**Journal Requirements:**

This paper represents independent research funded by the National Institute for Health and Care Research (NIHR) Biomedical Research Centre at South London and Maudsley NHS Foundation Trust in partnership with King’s College London. The views expressed are those of the author(s) and not necessarily those of the NHS, the NIHR or the Department of Health and Social Care. We acknowledge and are very grateful for the support of the Maudsley Biomedical Research Centre’s Young Person’s Mental Health Advisory Group, and the Maudsley Biomedical Research Centre’s Service User Advisory Group.

Professor Young’s independent research is funded by the National Institute for Health and Care Research (NIHR) Maudsley Biomedical Research Centre at South London and Maudsley NHS Foundation Trust and King's College London. The views expressed are those of the author(s) and not necessarily those of the NIHR or the Department of Health and Social Care.

MH reports funding from NIHR. AY reports funding from Flow Neuroscience, Novartis, Roche, Janssen, Takeda, Noema pharma, Compass, Astrazenaca, Boehringer Ingelheim, Eli Lilly, LivaNova, Lundbeck, Sunovion, Servier, Livanova, Janssen, Allegan, Bionomics, Sumitomo Dainippon Pharma, Sage, Neurocentrx, NIMH, CIHR, NARSAD, Stanley Medical Research Institute, MRC, Wellcome Trust, Royal College of Physicians, BMA, MSFHR, EU Horizon 2020, NIHR. No other authors reported competing interests. 

We note that you received funding from a commercial source: 

Reviewers' comments:

Reviewer's Responses to Questions

**Comments to the Author**

1. Is the manuscript technically sound, and do the data support the conclusions?

Reviewer #1: Partly

Reviewer #2: Partly

2. Has the statistical analysis been performed appropriately and rigorously? 

Reviewer #1: N/A

Reviewer #2: I Don't Know

3. Have the authors made all data underlying the findings in their manuscript fully available?

Reviewer #1: No

Reviewer #2: Yes

4. Is the manuscript presented in an intelligible fashion and written in standard English?

Reviewer #1: Yes

Reviewer #2: Yes

5. Review Comments to the Author

**Reviewer #1: ** The subjects were recruited who had formal recorded clinical diagnosis of psychotic major depression in the past three years. It appears the 38 former patients (“service users”) were identified and 10 agreed to participate. The study implies that PMD had been “diagnosed late” but we know nothing about that; we're all 38 former patients diagnosed late, or just the 10 who had been interviewed. Were there clinical differences between the 28 non participants and the 10 who agreed to participate? At what point did the true diagnosis of PMD become apparent and what contributed to the appropriate diagnosis?

The authors do a sterling job of representing the difficulties patients experience dealing with mental healthcare providers and also in expressing themselves when discussing intimate events in their lives with virtual strangers. There is no question that clinicians need to be more empathetic and that they ought to communicate more effectively with patience who come from a variety of different backgrounds and have different levels of understanding of their own psychology. For that reason alone, this paper is eminently publishable.

The authors also do an excellent job of describing the various reasons why a diagnosis might be missed but it would certainly be useful to understand how the diagnosis finally came to be made. Do the clinical records record the patient's level of understanding, the clinicians appreciation of the accuracy of the patients representations, and the difficulties that clinicians might have in generating in accurate clinical history? Is it true that clinicians failed to query the presence of psychotic symptoms? Patients are ordinarily given symptom checklists or rating scales to respond to and most of those ask about psychotic symptoms. Not all of them do, however. Rating scales and symptom checklists, however, are good ways to make sure all of the bases are covered when one is taking a medical history and generating a proper diagnosis.

The authors correctly identify memory problems that patients may have, especially when the clinical encounters may have occurred three years earlier. They might also emphasize the cognitive disorganization that frequently accompanies severe depression, especially major depression with psychosis. One doesn't wish to discount the accuracy of patients’ reports but human memory is inevitably fallible and it is possible, even in the context of semi structured interview, for the interviewers presuppositions to influence patients’ responses during a research interview.

My only important recommendation to the authors is to inform readers how the proper diagnosis of PMDD was ultimately made and to review patients’ commentaries in that light.

**Reviewer #2: ** Service user perspectives and experiences of their diagnosis of psychotic major depression: a qualitative study

The topic is interesting, but

Why u choice this topic

In the introduction part, you should only focused about psychotic major depression

What tools do you used to assess psychotic major depression?

What is the difference between major depression disorder with psychotic features, schizoaffective disorder and psychotic major depression?

Is schizoaffective disorder or not?

To Dx a patient with psychotic major depression is based- on DSM-5 or not, if not by what criteria will be diagnosis with psychotic major depression?

Is new finding on the management of major depression disorder with psychotic features, psychotic major depression and schizoaffective disorder? If yes what is new finding on the management of psychotic major depression?

In conclusions part: Psychotic major depression should be actively considered as a differential diagnosis by healthcare professionals when assessing an individual for unipolar depression, it is not understand, to me my 1st DDX will be schizoaffective disorder or bipolar 1 disorder with psychotic feature.

References, there problems on references examples Hales [11] suggests that because patients recognise that their thought patterns are unusual, they may attempt to conceal their symptoms from others…..

Which tools were used to assess barriers, and stigma…

Study population, study unit

What sampling methods and technique were used to asses

Why only 10 people participated in the interviews?

Service user frustration with diagnostic instability and lack of certainty, what does it mean

In the Discussion part, there is no references, so put references and discuss with different studies

How they know whether the diagnoses were psychotic major depression or not?

What does it mean medical malpractice

Why you exclude individuals above 64 years old from the study

6. PLOS authors have the option to publish the peer review history of their article (what does this mean? ). If published, this will include your full peer review and any attached files.

**Do you want your identity to be public for this peer review?** For information about this choice, including consent withdrawal, please see our Privacy Policy .

Reviewer #1: **Yes: ** C Thomas Gualtieri MD

Reviewer #2: **Yes: ** Yes, my full name you can appear in the published peer review

---

## [Author Response · Author response to Decision Letter 1]

14 Jan 2025

See response to reviewers document

---

## [Decision Letter · Decision Letter 1]

31 Mar 2025

Service user perspectives and experiences of their diagnosis of psychotic major depression: a qualitative study

PONE-D-24-19547R1

Dear Dr. Heslin,

We’re pleased to inform you that your manuscript has been judged scientifically suitable for publication and will be formally accepted for publication once it meets all outstanding technical requirements.

Kind regards,

Valentina Baldini

Academic Editor

PLOS ONE

Additional Editor Comments (optional):

Dear Authors,

We are pleased to inform you that your manuscript, entitled Service user perspectives and experiences of their diagnosis of psychotic major depression: a qualitative study , has been accepted for publication in PLOS ONE. We appreciate the time and effort you have dedicated to this research and acknowledge its contribution to the field.

Your manuscript has undergone thorough peer review, and we commend your diligence in addressing the reviewers' comments and improving the clarity, rigor, and impact of your work. The revisions and responses you provided have satisfactorily addressed the concerns raised, ensuring that the study meets the high standards of scientific integrity and reproducibility upheld by PLOS ONE.

At this stage, your manuscript will proceed to the production phase, where it will be formatted and prepared for online publication. You will receive further communication regarding proofs and any necessary editorial adjustments. Please ensure that all co-authors verify their final details and disclosures before the article is officially published.

We congratulate you on this achievement and look forward to seeing the impact of your research within the academic community. Thank you for choosing PLOS ONE as the platform to disseminate your work.

Best regards,

Valentina Baldini

Academic Editor,

PLOS ONE

Reviewers' comments:

Reviewer's Responses to Questions

**Comments to the Author**

1. If the authors have adequately addressed your comments raised in a previous round of review and you feel that this manuscript is now acceptable for publication, you may indicate that here to bypass the “Comments to the Author” section, enter your conflict of interest statement in the “Confidential to Editor” section, and submit your "Accept" recommendation.

Reviewer #1: All comments have been addressed

2. Is the manuscript technically sound, and do the data support the conclusions?

Reviewer #1: Yes

3. Has the statistical analysis been performed appropriately and rigorously? 

Reviewer #1: N/A

4. Have the authors made all data underlying the findings in their manuscript fully available?

Reviewer #1: Yes

5. Is the manuscript presented in an intelligible fashion and written in standard English?

Reviewer #1: Yes

6. Review Comments to the Author

Reviewer #1: (No Response)

7. PLOS authors have the option to publish the peer review history of their article (what does this mean? ). If published, this will include your full peer review and any attached files.

**Do you want your identity to be public for this peer review?** For information about this choice, including consent withdrawal, please see our Privacy Policy .

Reviewer #1: **Yes: ** C T Gualtieri

---

## [Editor Report · Acceptance letter]

PONE-D-24-19547R1

PLOS ONE

Dear Dr. Heslin,

I'm pleased to inform you that your manuscript has been deemed suitable for publication in PLOS ONE. Congratulations! Your manuscript is now being handed over to our production team.

Kind regards,

on behalf of

Dr. PLOS Manuscript Reassignment

Staff Editor

PLOS ONE